# Training and Inference with Integers in Deep Neural Networks

**Shuang Wu**[1]**, Guoqi Li**[1]**, Feng Chen**[2]**, Luping Shi**[1]
[1]Department of Precision Instrument
[2]Department of Automation
Center for Brain Inspired Computing Research
Beijing Innovation Center for Future Chip
Tsinghua University
{lpshi,chenfeng}@mail.tsinghua.edu.cn

## Abstract

Researches on deep neural networks with discrete parameters and their deployment in embedded systems have been active and promising topics. Although previous works have successfully reduced precision in inference, transferring both training and inference processes to low-bitwidth integers has not been demonstrated simultaneously. In this work, we develop a new method termed as "WAGE" to discretize both training and inference, where weights (W), activations (A), gradients (G) and errors (E) among layers are shifted and linearly constrained to low-bitwidth integers. To perform pure discrete dataflow for fixed-point devices, we further replace batch normalization by a constant scaling layer and simplify other components that are arduous for integer implementation. Improved accuracies can be obtained on multiple datasets, which indicates that WAGE somehow acts as a type of regularization. Empirically, we demonstrate the potential to deploy training in hardware systems such as integer-based deep learning accelerators and neuromorphic chips with comparable accuracy and higher energy efficiency, which is crucial to future AI applications in variable scenarios with transfer and continual learning demands.

## 1 Introduction

Recently deep neural networks (DNNs) are being widely used for numerous AI applications (Krizhevsky et al., 2012; Hinton et al., 2012; Silver et al., 2016). Depending on the massive tunable parameters, DNNs are considered to have powerful multi-level feature extraction and representation abilities. However, training DNNs needs energy-intensive devices such as GPU and CPU with high precision (float32) processing units and abundant memory, which has greatly challenged their extensive applications for portable devices. In addition, a state-of-art network often has far more weights and effective capacity to shatter all training samples (Zhang et al., 2016), leading to overfitting easily.

As a result, there is much interest in reducing the size of network during inference (Hubara et al., 2016; Rastegari et al., 2016; Li et al., 2016), as well as dedicated hardware for commercial solutions (Jouppi et al., 2017; Chen et al., 2017; Shi et al., 2015). Due to the accumulation in stochastic gradient descent (SGD) optimization, the precision demand for training is usually higher than inference (Hubara et al., 2016; Li et al., 2017). Therefore, most of the existing techniques only focus on the deployment of a well-trained compressed network, while still keeping high precision and computational complexity during training. In this work, we address this problem as how to process both training and inference with low-bitwidth integers, which is essential for implementing DNNs in dedicated hardware. To this end, two fundamental issues are addressed for discretely training DNNs: i) how to quantize all the operands and operations, and ii) how many bits or states are needed for SGD computation and accumulation.

With respect to the issues, we propose a framework termed as "WAGE" that constrains weights (W), activations (A), gradients (G) and errors (E) among all layers to low-bitwidth integers in both training and inference. Firstly, for operands, linear mapping and orientation-preserved shifting are applied

to achieve ternary weights, 8-bit integers for activations and gradients accumulation. Secondly, for operations, batch normalization (Ioffe & Szegedy, 2015) is replaced by a constant scaling factor. Other techniques for fine-tuning such as SGD optimizer with momentum and L2 regularization are simplified or abandoned with little performance degradation. Considering the overall bidirectional propagation, we completely streamline inference into accumulate-compare cycles and training into low-bitwidth multiply-accumulate (MAC) cycles with alignment operations, respectively.

We heuristically explore the bitwidth requirements of integers for error computation and gradient accumulation, which have rarely been discussed in previous works. Experiments indicate that it is the relative values (orientations) rather than absolute values (orders of magnitude) in error that guides previous layers to converge. Moreover, small values have negligible effects on previous orientations though propagated layer by layer, which can be partially discarded in quantization. We leverage these phenomena and use an orientation-preserved shifting operation to constrain errors. As for the gradient accumulation, though weights are quantized to ternary values in inference, a relatively higher bitwidth is indispensable to store and accumulate gradient updates.

The proposed framework is evaluated on MNIST, CIFAR10, SVHN, ImageNet datasets. Comparing to those who only discretize weights and activations at inference time, it has comparable accuracy and can further alleviate overfitting, indicating some type of regularization. WAGE produces pure bidirectional low-precision integer dataflow for DNNs, which can be applied for training and inference in dedicated hardware neatly. We publish the code on GitHub[1].

## 2 RELATED WORK

We mainly focus on reducing precision of operands and operations in both training and inference. Orthogonal and complementary techniques for reducing complexity like network compression, pruning (Han et al., 2015; Zhou et al., 2017) and compact architectures (Howard et al., 2017) are impressively efficient but outside the scope this paper.

**Weight and activation** Courbariaux et al. (2015); Hubara et al. (2016) propose methods to train DNNs with binary weights (BC) and activations (BNN) successively. They add noises to weights and activations as a form of regularization but real-valued gradients are accumulated in real-valued variables, suggesting that high precision accumulation is likely required for SGD optimization. XNOR-Net (Rastegari et al., 2016) has a filter-wise scaling factor for weights to improve the performance. Convolutions in XNOR-Net can be implemented efficiently using XNOR logical units and bit-count operations. However, these floating-point factors are calculated simultaneously during training, which generally aggravates the training effort. In TWN (Li et al., 2016) and TTQ (Zhu et al., 2016) two symmetric thresholds are introduced to constrain the weights to be ternary-valued: $\{+1, 0, -1\}$. They claimed a tradeoff between model complexity and expressive ability.

**Gradient computation and accumulation** DoReFa-Net (Zhou et al., 2016) quantizes gradients to low-bitwidth floating-point numbers with discrete states in the backward pass. TernGrad (Wen et al., 2017) quantizes gradient updates to ternary values to reduce the overhead of gradient synchronization in distributed training. Nevertheless, weights in DoReFa-Net and TernGrad are stored and updated with float32 during training like previous works. Besides, the quantization of batch normalization and its derivative is ignored. Thus, the overall computation graph for the training process is still presented with float32 and more complex with external quantization. Generally, it is difficult to apply DoReFa-Net training in an integer-based hardware directly, but it shows potential for exploring high-dimensional discrete spaces with discrete gradient descent directions.

## 3 WAGE QUANTIZATION

The main idea of WAGE quantization is to constrain four operands to low-bitwidth integers: weight $W$ and activation $a$ in inference, error $e$ and gradient $g$ in backpropagation training, see Figure 1. We extend the original definition of errors to multi-layer: error $e$ is the gradient of activation $a$ for the perspective of each convolution or fully-connected layer, while gradient $g$ particularly refers to the gradient accumulation of weight $W$. Considering the $i$-th layer of a feed-forward network, we

---

[1]https://github.com/boluoweifenda/WAGE

have:

$$e^i = \frac{\partial \mathcal{L}}{\partial a^i}, \; g^i = \frac{\partial \mathcal{L}}{\partial W^i} \tag{1}$$

where $\mathcal{L}$ is the loss function. We separate these two terms that are mixed up in most existing schemes. The gradient of weight $g$ and the gradient of activation $e$ flow to different paths in each layer, which is a fork both in inference and in backward training and generally acts as node of MAC operations.

For the forward propagation in the $i$-th layer, assuming that weights are stored and accumulated with $k_G$-bit integers, then numerous works strive for a better quantization function $Q_W(\cdot)$ that maps higher precision weights to their $k_W$-bit reflections, for example, $[-0.9, 0.1, 0.7]$ to $[-1, 0, 1]$. Although weights are accumulated with high precision like float32, the deployment of the reflections in dedicated hardware are much more memory efficient after training. Activations are quantized with function $Q_A(\cdot)$ to $k_A$ bits to align the increased bitwidth caused by MACs. Weights and activations are discretized to even binary values in previous works, then MACs degrade into logical and bit-count operations that are extremely efficient (Rastegari et al., 2016).

For the backward propagation in the $i$-th layer, the gradients of activations and weights are calculated by the derivatives of MACs that are generally considered to be in 16-bit floating-point precision at least. As illustrated in Figure 1, the MACs between $k_A$-bit inputs and $k_W$-bit weights will increase the bitwidth of outputs to $[k_A + k_W - 1]$ in signed integer representation, and the similar broadening happens to errors $e$ as well. In consideration of training with only low-bitwidth integers, we propose additional functions $Q_E(\cdot)$ and $Q_G(\cdot)$ to constrain bitwidth of $e$ and $g$ to $k_E$ bits and $k_G$ bits, respectively. In general, where there is a MAC operation, there are quantization operators named $Q_W(\cdot)$, $Q_A(\cdot)$, $Q_G(\cdot)$ and $Q_E(\cdot)$ in inference and backpropagation.

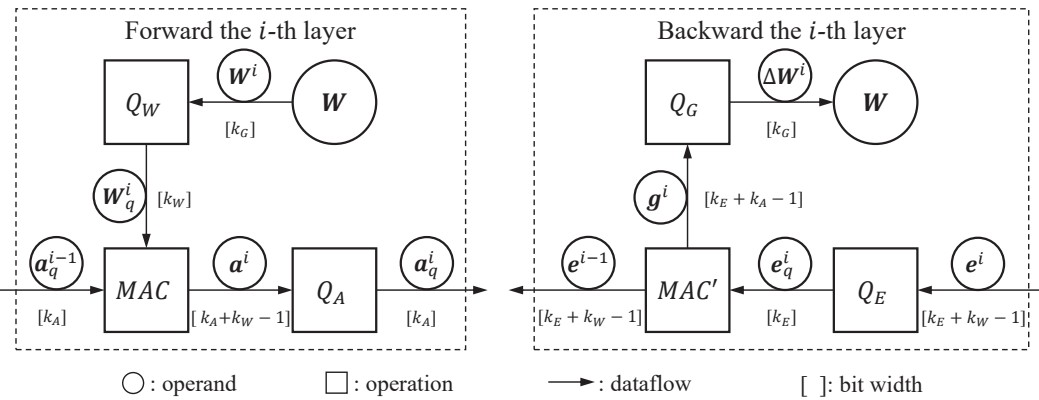

Figure 1: Four operators $Q_W(\cdot)$, $Q_A(\cdot)$, $Q_G(\cdot)$, $Q_E(\cdot)$ added in WAGE computation dataflow to reduce precision, bitwidth of signed integers are below or on the right of arrows, activations are included in MAC for concision.

## 3.1 Shift-based Linear Mapping and Stochastic Rounding

In WAGE quantization, we adopt a linear mapping with $k$-bit integers for simplicity, where continuous and unbounded values are discretized with uniform distance $\sigma$:

$$\sigma(k) = 2^{1-k}, k \in \mathbb{N}_+ \tag{2}$$

Then the basic quantization function that converts a floating-point number $x$ to its $k$-bitwidth signed integer representation can be formulated as:

$$Q(x, k) = Clip\left\{ \sigma(k) \cdot round\left[\frac{x}{\sigma(k)}\right], -1 + \sigma(k), 1 - \sigma(k) \right\} \tag{3}$$

where $round$ approximates continuous values to their nearest discrete states. $Clip$ is the saturation function that clips unbounded values to $[-1 + \sigma, 1 - \sigma]$, where the negative maximum value $-1$ is

removed to maintain symmetry. For example, $Q(x, 2)$ quantizes $\{-1, 0.2, 0.6\}$ to $\{-0.5, 0, 0.5\}$. Equation 3 is merely used for simulation in floating-point hardware like GPU, whereas in a fixed-point device, quantization and saturation is satisfied automatically.

Before applying linear mapping in some operands (e.g., error), we introduce an additional monolithic scaling factor for shifting values distribution to an appropriate order of magnitude, otherwise values will be all saturated or cleared by Equation 3. The scaling factor is calculated by $Shift$ function and then divided in later steps:

$$Shift(x) = 2^{round(\log_2 x)} \tag{4}$$

Finally, we propose stochastic rounding to substitute small and real-valued updates for gradient accumulation in training. Section 3.3.4 will detail the implementation of operator $Q_G(\cdot)$, where high bitwidth gradients are constrained to $k_G$-bit integers stochastically by a 16-bit random number generator. Figure 2 summarizes quantization methods used in WAGE.

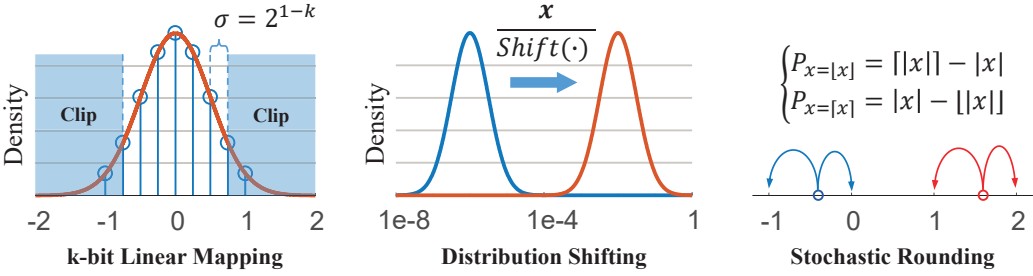

Figure 2: Quantization methods used in WAGE. The notation $P$, $\boldsymbol{x}$, $\lfloor \cdot \rfloor$ and $\lceil \cdot \rceil$ denotes probability, vector, $floor$ and $ceil$, respectively. $Shift(\cdot)$ refers to Equation 4 with a certain argument.

## 3.2 WEIGHT INITIALIZATION

In previous works, weights are binarized directly by $sgn$ function or ternarized by threshold parameters calculated during training. However, BNN fails to converge without batch normalization because weight values $\pm 1$ are rather big for a typical DNN. Batch normalization not only efficiently avoids the problem of exploding and vanishing gradients, but also alleviates the demand for proper initialization. However, normalizing outputs for each layer and computing their gradients are quite complex without floating point unit (FPU). Besides, the moving averages of batch outputs occupy external memory. BNN shows a shift-based variation of batch normalization but it is hard to transform all of the elements to the fixed-point representations. As a result, weights should be cautiously initialized in this work where batch normalization is simplified to a constant scaling layer. A modified initialization method based on MSRA (He et al., 2015) can be formulated as:

$$\boldsymbol{W} \sim U(-L, +L), L = max\{\sqrt{6/n_{in}}, L_{min}\}, L_{min} = \beta\sigma \tag{5}$$

where $n_{in}$ is the layer fan-in number, and the original limit $\sqrt{6/n_{in}}$ in MSRA is calculated to keep same variance between inputs and outputs of the same layer theoretically. The additional limit $L_{min}$ is a minimum value that the uniform distribution $U$ should reach, and $\beta$ is a constant greater than 1 to create overlaps between minimum step size $\sigma$ and maximum value $L$. In case of $k_W$-bit linear mapping, if weights $\boldsymbol{W}$ are quantized directly with original limits, we will get all-zero tensors when bitwidth $k_W$ is small enough, e.g., 4, or fan-in $n_{in}$ is wide enough, where initialized weights may never reach the minimum step $\sigma$ presented by fixed-point integers. So $L_{min}$ ensures that weights can go beyond $\sigma$ and quantized to non-zero values after $Q_W(\cdot)$ when initialized randomly.

## 3.3 QUANTIZATION DETAILS

### 3.3.1 WEIGHT $Q_W(\cdot)$

The modified initialization in Equation 5 will amplify weights holistically and guarantee their proper distribution, then $\boldsymbol{W}$ is quantized directly with Equation 3:

$$\boldsymbol{W}_q = Q_W(\boldsymbol{W}) = Q(\boldsymbol{W}, k_W) \tag{6}$$

It should be noted that the variance of weights is scaled compared to the original limit, which will cause exploding of network's outputs. To alleviate the amplification effect, XNOR-Net proposed a filter-wise scaling factor calculated continuously with full precision. In consideration of integer implementation, we introduce a *layer-wise shift-based* scaling factor $\alpha$ to attenuate the amplification effect:

$$\alpha = max\{Shift(L_{min}/L), 1\} \tag{7}$$

where $\alpha$ is a pre-defined constant for each layer determined by the network structure. The modified initialization and attenuation factor $\alpha$ together approximates floating-point weights to their integer representations, except that $\alpha$ takes effect after activations to maintain precision of weights presented by $k_W$-bit integers.

### 3.3.2 Activation $Q_A(\cdot)$

As stated above, the bitwidth of operands increases after MACs. Then a typical CNN is usually followed with pooling, normalization and activation. Average pooling is avoided because mean operations will increase precision demand. Besides, we hypothesize that batch outputs of each hidden layer approximately have zero-mean, then batch normalization degenerates into to a scaling layer where trainable and batch-calculated scaling parameters are replaced by $\alpha$ mentioned in Equation 7. If activations are presented in $k_A$ bits, the overall quantization of activations can be formulated as:

$$\boldsymbol{a}_q = Q_A(\boldsymbol{a}) = Q(\boldsymbol{a}/\alpha, k_A) \tag{8}$$

### 3.3.3 Error $Q_E(\cdot)$

Errors $\boldsymbol{e}$ are calculated layer by layer using the chain rule during training. Although the computation graph of backpropagation is similar to the inference, the inputs are the gradients of $\mathcal{L}$, which are relatively small compared to actual inputs for networks. More importantly, the errors are unbounded and might have significantly larger ranges than that of activations, e.g., $[10^{-9}, 10^{-4}]$. DoReFa-Net first applies an affine transform on $\boldsymbol{e}$ to map them into $[-1, 1]$, and then inverts the transform after quantization. Thus, the quantized $\boldsymbol{e}$ are still presented as float32 numbers with discrete states and mostly small values.

However, experiments uncover that it is the orientations rather than orders of magnitude in errors that guides previous layers to converge, then the inverse transformation after quantization in DoReFa-Net is no longer needed. The orientation-only preservation prompts us to propagate errors with integer thoroughly, where error distribution is firstly scaled into $[-\sqrt{2}, +\sqrt{2}]$ by dividing a shift factor as shown in Figure 2 and then quantized by $Q(\boldsymbol{e}, k_E)$:

$$\boldsymbol{e}_q = Q_E(\boldsymbol{e}) = Q(\boldsymbol{e}/Shift(max\{|\boldsymbol{e}|\}), k_E) \tag{9}$$

where $max\{|\boldsymbol{e}|\}$ extracts the layer-wise maximum absolute value among all elements in error $\boldsymbol{e}$, multi-channel for convolution and multi-sample for batch training. The quantization of error discards large proportion of values smaller than $\sigma$, we will discuss the influence on accuracy later.

### 3.3.4 Gradient $Q_G(\cdot)$

Since we only preserve relative values of error after shifting, the gradient updates $\boldsymbol{g}$ derived from MACs between backward errors $\boldsymbol{e}$ and forward activations $\boldsymbol{a}$ are shifted consequently. We first rescale gradients $\boldsymbol{g}$ with another scaling factor and then bring in shift-based learning rate $\eta$:

$$\boldsymbol{g}_s = \eta \cdot \boldsymbol{g}/Shift(max\{|\boldsymbol{g}|\}) \tag{10}$$

where $\eta$ is an integer power of 2. The shifted gradients $\boldsymbol{g}_s$ represent for minimum step numbers and directions for updating weights. If weights are stored with $k_G$-bit numbers, the minimum step of modification will be $\pm 1$ for integers and $\pm\sigma(k_G)$ for floating-point values, respectively. The implement of learning rate $\eta$ here is quite different from that in a vanilla DNN based on float32. In WAGE, there only remain directions for weights to change and the step sizes are integer multiples of minimum step $\sigma$. Shifted gradients $\boldsymbol{g}_s$ may get greater than 1 if $\eta$ is 2 or bigger to accelerate training at the beginning, or smaller than 0.5 during latter half of training when learning rate decay is usually applied. As illustrated in Figure 2, to substitute accumulation of small gradients in latter

case, we separate $\boldsymbol{g}_s$ into integer parts and decimal parts, then use a 16-bit random number generator to constrain high bitwidth $\boldsymbol{g}_s$ to $k_G$-bit integers stochastically:

$$\Delta \boldsymbol{W} = Q_G(\boldsymbol{g}) = \sigma(k_G) \cdot sgn(\boldsymbol{g}_s) \cdot \left\{ \lfloor |\boldsymbol{g}_s| \rfloor + Bernoulli(\, |\boldsymbol{g}_s| - \lfloor |\boldsymbol{g}_s| \rfloor \,) \right\} \tag{11}$$

where $Bernoulli$ (Zhou et al., 2016) stochastically samples decimal parts to either 0 or 1. With proper setting of $k_G$, quantization of gradients will restrict the minimum step size, which may avoid local minimum and overfitting. Furthermore, the gradients will be ternary values when $\eta$ is not greater than 1, which reduces communication costs for distributed training (Wen et al., 2017). At last, weights $\boldsymbol{W}$ might exceed the range $[-1 + \sigma, 1 - \sigma]$ presented by $k_G$-bit integers after updating with discrete increments $\Delta \boldsymbol{W}$. So $Clip$ function is indispensable to saturate and make sure there are only $2^{k_G-1} - 1$ states for weights accumulation. In case of the $t$-th iteration, we have:

$$\boldsymbol{W}_{t+1} = Clip\left\{\boldsymbol{W}_t - \Delta \boldsymbol{W}_t, -1 + \sigma(k_G), 1 - \sigma(k_G)\right\} \tag{12}$$

## 3.4 Miscellaneous

From the above, we have illustrated our quantization methods for weights, activations, gradients and errors. See Algorithm 1 for the detailed computation graph. There remain some issues to specify in an overall training process with only integers.

Gradient descent optimizer like Momentum, RMSProp and Adam contains at least one copy of gradient updates $\Delta \boldsymbol{W}$ or their moving average, doubling memory consumption for weights during training, which is partially equivalent to use bigger $k_G$. Since the weight updates $\Delta \boldsymbol{W}$ are quantized to integer multiple of $\sigma$ and scaled by $\eta$, we adopt pure mini-batch SGD without any form of momentum or adaptive learning rate to show the potential of reducing storage demands.

Although L2 regularization works quite well for many large-scale DNNs where overfitting occurs commonly, WAGE removes small values in Equation 3 and introduces randomness in Equation 11, acting as certain types of regularization and can get comparable accuracy in later experiments. Thus, we remain L2 weight decay and dropout as supplementary regularization methods.

The *Softmax* layer and cross-entropy criterion are widely adopted in classification tasks but the calculation of $e^x$ can hardly be applied in low-bitwidth linear mapping occasions. For tasks with small number of categories, we avoid *Softmax* layer and apply mean-square-error criterion but omit mean operation to form a sum-square-error (SSE) criterion since shifted errors will get the same values in Equation 9.

## 4 Experiments

In this section, we set W-A-G-E bits to 2-8-8-8 as default for all layers in a CNN or MLP. The bitwidth $k_W$ is 2 for ternary weights, which implies that there are no multiplications during inference. Constant parameter $\beta$ is 1.5 to make equal probabilities for ternary weights when initialized randomly. Activations and errors should be of the same bitwidth since computation graph of backpropagation is similar to inference and might be applied in the same partition of hardware or memristor array (Sheridan et al., 2017). Although XNOR-Net achieves 1-bit activations, reducing errors to 4 or less bits dramatically degenerates accuracies in our tests, so the bitwidth $k_A$ and $k_E$ are increased to 8 simultaneously. Weights are stored with 8-bit integers during training and ternarized by two constant symmetrical thresholds during inference. We first build the computation graph for a vanilla network, then insert quantization nodes in forward propagation and override gradients in backward propagation for each layer on Tensorflow (Abadi et al., 2016). Our method is evaluated on MNIST, SVHN, CIFAR10 and ILSVRC12 (Russakovsky et al., 2015) and Table 1 shows the comparison results.

## 4.1 Implement Details

**MNIST:** A variation of LeNet-5 (LeCun et al., 1998) with 32C5-MP2-64C5-MP2-512FC-10SSE is adopted. The input grayscale images are regarded as activations and quantized by Equation 8 where $\alpha$ equals to 1. The learning rate $\eta$ in WAGE remains as 1 for the whole 100 epochs. We report average accuracy of 10 runs on the test set.

**SVHN & CIFAR10:** We use a VGG-like network (Simonyan & Zisserman, 2014) with $2\times$(128C3)-MP2-$2\times$(256C3)-MP2-$2\times$(512C3)-MP2-1024FC-10SSE. For CIFAR10 dataset, we follow the data augmentation in Lee et al. (2015) for training: 4 pixels are padded on each side, and a $32\times32$ patch is randomly cropped from the padded image or its horizontal flip. For testing, only single view of the original $32\times32$ image is evaluated. The model is trained with mini-batch size of 128 and totally 300 epochs. Learning rate $\eta$ is set to 8 and divided by 8 at epoch 200 and epoch 250. The original images are scaled and biased to the range of $[-1, +1]$ for 8-bit integer activation representation. As for SVHN dataset, we leave out randomly flip augmentation and reduce training epochs to 40 since it is a rather big dataset. The error rate is evaluated in the same way as MNIST.

**ImageNet:** WAGE framework is evaluated on ILSVRC12 dataset with AlexNet (Krizhevsky et al., 2012) model but removes dropout and local response normalization layers. Images are firstly resized to $256\times256$ then randomly cropped to $224\times224$ and horizontally flipped, followed by bias subtraction as CIFAR10. For testing, the single center crop in validation set is evaluated. Since ImageNet task is much difficult than CIFAR10 and has 1000 categories, it is hard to converge when applying SSE or hinge loss criterion in WAGE, so we add *Softmax* and remove quantizations in the last layer for fear of severe accuracy drop (Tang et al., 2017). The model is trained with mini-batch size of 256 and totally 70 epochs. Learning rate $\eta$ is set to 4 and divided by 8 at epoch 60 and epoch 65.

Table 1: Test or validation error rates (%) in previous works and WAGE on multiple datasets. Opt denotes gradient descent optimizer and withM means SGD with momentum, BN represents for batch normalization and 32 bits refers to float32, ImageNet top-k format: top1/top5.

| Method | $k_W$ | $k_A$ | $k_G$ | $k_E$ | Opt | BN | MNIST | SVHN | CIFAR10 | ImageNet |
|---|---|---|---|---|---|---|---|---|---|---|
| BC | 1 | 32 | 32 | 32 | Adam | ✓ | 1.29 | 2.30 | 9.90 | - |
| BNN | 1 | 1 | 32 | 32 | Adam | ✓ | 0.96 | 2.53 | 10.15 | - |
| BWN[1] | 1 | 32 | 32 | 32 | withM | ✓ | - | - | - | 43.2/20.6 |
| XNOR | 1 | 1 | 32 | 32 | Adam | ✓ | - | - | - | 55.8/30.8 |
| TWN | 2 | 32 | 32 | 32 | withM | ✓ | 0.65 | - | 7.44 | 34.7/13.8 |
| TTQ | 2 | 32 | 32 | 32 | Adam | ✓ | - | - | 6.44 | 42.5/20.3 |
| DoReFa[2] | 8 | 8 | 32 | 8 | Adam | ✓ | - | 2.30 | - | 47.0/ - |
| TernGrad[3] | 32 | 32 | 2 | 32 | Adam | ✓ | - | - | 14.36 | 42.4/19.5 |
| WAGE | 2 | 8 | 8 | 8 | SGD | ✗ | **0.40** | **1.92** | 6.78 | 51.6/27.8 |

## 4.2 Training Curves and Regularization

We further compare WAGE variations and a vanilla CNN on CIFAR10. The vanilla CNN has the same VGG-like architecture described above except that none quantization of any operand or operation is applied. We add batch normalization in each layer and *Softmax* for the last layer, replace SSE with cross-entropy criterion, and then use a L2 weight decay of 1e-4 and momentum of 0.9 for training. The learning rate is set to 0.1 and divided by 10 at epoch 200 and epoch 250. For variations of WAGE, pattern 28ff has no quantization nodes in backpropagation. Although the 28ff pattern has the same optimizer and learning rate annealing method as the vanilla pattern, we find that weight updates are decreased by the rescale factor $\alpha$ in Equation 7. Therefore, the learning rate for 28ff is amplified and tuned, which reduces the error rate by 3%. Figure 3 shows the training curves of three counterparts. It can be seen that the 2888 pattern has comparable convergence rate to the vanilla CNN, better accuracy than those who only discretize weights and activations in inference time, though slightly more volatile. The discretization of backpropagation somehow acts as another type of regularization and have significant error rate drop when decreasing learning rate $\eta$.

---

[1]BWN is the counterpart of XNOR that only quantizes weights

[2]They use floating-point presentation for errors

[3]Only for worker-to-server communication in distributed training, weights are accumulated with float32

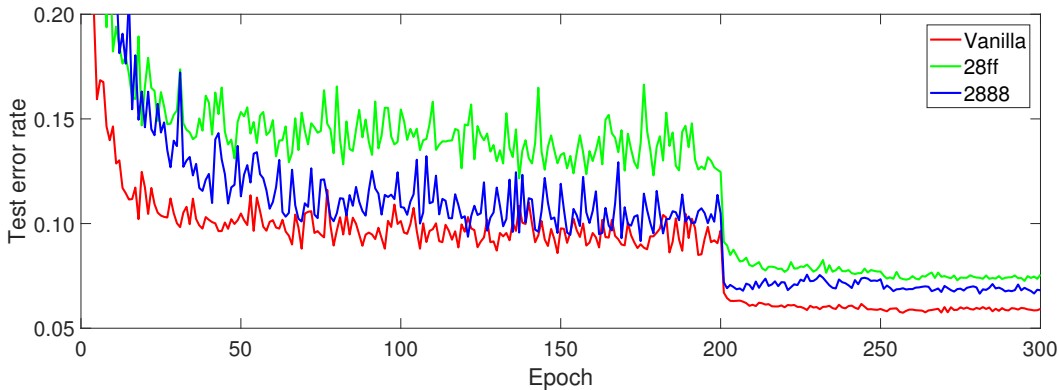

Figure 3: Training curves of WAGE variations and a vanilla CNN on CIFAR10.

### 4.3 BITWIDTH OF ERRORS

The bitwidth $k_E$ is set to 8 as default in previous experiments. To further explore a proper bitwidth and its truncated boundary, we firstly export errors from vanilla CNN for CIFAR10 after 100 training epochs. The histogram of errors in the last convolution layer among 128 mini-batch data is shown in Figure 4. It is obvious that errors approximately obey logarithmic normal distribution where values are relatively small and have significantly large range. When quantized with $k_E$-bit integers, a proper window function should be chosen to truncate the distribution while retaining the approximate orientations for backpropagation. For more details about the layerwise histograms of all W, A, G, E operands, see Figure 5.

Firstly, the upper (right) boundary is immobilized to the maximum absolute value among all elements in errors as described in Equation 9. Then the left boundary will be based on the bitwidth $k_E$. We conduct a series of experiments for $k_E$ ranging from 4 to 15. The boxplot in Figure 4 indicates that 4-8 bits of errors represented by integers are enough for CIFAR10 classification task. Bitwidth 8 is chosen as default to match the 8-bit image color levels and most operands in the micro control unit (MCU). The histogram of errors in the same layer of WAGE-2888 shows that after being shifted and quantized layer by layer, the distribution of errors reshapes and mostly aggregates into truncated window. Thus, most information for orientations is retained. Besides, the smaller values in errors have negligible effects on previous orientations though accumulated layer by layer, which are partially discarded in quantization.

Since the width of the window has been optimized, we left-shift the window with factor $\gamma$ to explore its horizontal position. The right boundary can be formulated as $max\{|e|\}/\gamma$. Table 2 shows the effect of shifting errors: although large values are in the minority, they play critical roles for backpropagation training while the majority with small values actually act as noises.

Table 2: Test error rates (%) on CIFAR10 when left-shift upper boundary with factor $\gamma$.

| $\gamma$ | 1 | 2 | 4 | 8 |
|---|---|---|---|---|
| error | 6.78 | 7.31 | 8.08 | 16.92 |

### 4.4 BITWIDTH OF GRADIENTS

The bitwidth $k_G$ is set to 8 as default in previous experiments. Although weights are propagated with ternary values in inference and achieve $16\times$ compression rate than float32 weights, they are saved and accumulated in a relatively higher bitwidth (8 bits) for backpropagation training. Therefore, the overall compression rate is only $4\times$. The inconsistent bitwidth between weight updates $k_G$ and their effects in inference $k_W$ provides indispensable buffer space. Otherwise, there might be too many weights changing their ternary values in each iteration, making training very slow and unstable. To

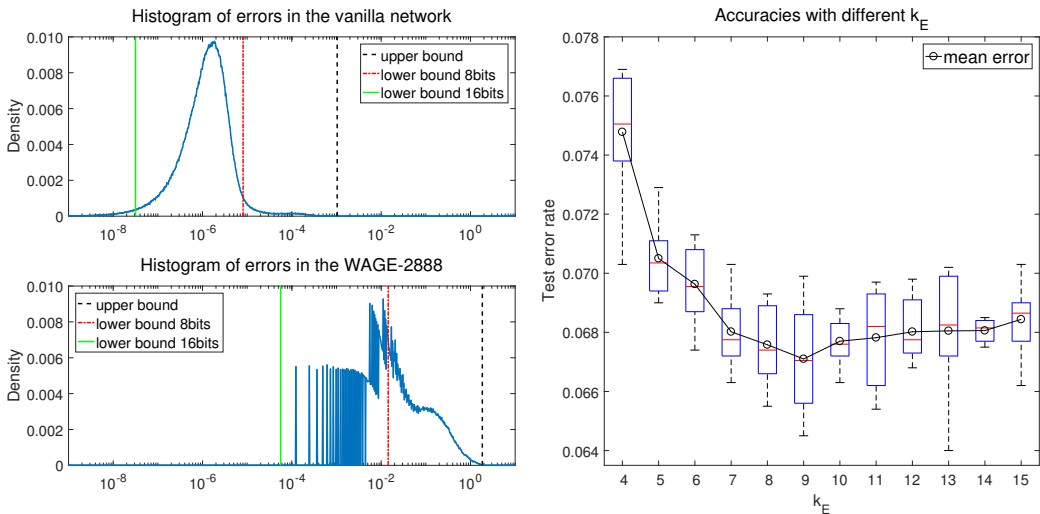

Figure 4: Left are histograms of errors $e$ for same layer in vanilla network and WAGE-2888 network. Upper boundaries are the $max\{|e|\}$ while lower boundaries are determined by the bitwidth $k_E$. The 10 run accuracies of different $k_E$ are shown on the right.

further explore a proper bitwidth for gradients, we use WAGE 2-8-8-8 in CIFAR10 as baseline and range $k_G$ from 2 to 12, the learning rate $\eta$ is divided by 2 every time the $k_G$ decreases 1 bit to keep approximately equal weights accumulation in large number of iterations. Results from Table 3 show the effect of $k_G$ and indicate the similar bitwidth requirement as previous experiments for $k_E$.

Table 3: Test error rates (%) on CIFAR10 with different $k_G$.

| $k_G$ | 2 | 3 | 4 | 5 | 6 | 7 | 8 | 9 | 10 | 11 | 12 |
|-------|-------|-------|-------|-------|-------|------|------|------|------|------|------|
| error | 54.22 | 51.57 | 28.22 | 18.01 | 11.48 | 7.61 | 6.78 | 6.63 | 6.43 | 6.55 | 6.57 |

For ImageNet implementation, we conduct six patterns to show bitwidth requirements: 2888 from Table 1, 288C for more accurate errors (12 bits), 28C8 for larger buffer space, 28f8 for none quantization of gradients, 28ff for errors and gradients in float32 as unlimited case and its BN counterpart. The accuracy of original AlexNet reproduction is reported as baseline. Learning rate $\eta$ is set to 64 and divided by 8 in 28C8 pattern, 0.01 and divided by 10 in 28f8, 28ff counterparts and vanilla AlexNet. We observe overfitting when increasing $k_G$ thus add L2 weight decay of 1e-4, 1e-4 and 5e-4 for 28f8, 28ff and 28ff-BN patterns, respectively. In table 4, the comparison between pattern 28C8 and 288C reveals that it might be more important to make more buffer space $k_G$ for gradient accumulation than to keep high-resolution orientation $k_E$. Besides, when it comes to ImageNet dataset, the gradient accumulation, i.e., the bit width of gradients ($k_G$) and batch normalization become more important (Li et al., 2017) since samples in training set are so variant.

To avoid external memory consumption of full-precision weights during training, Deng et al. (2018) achieved 1-bit weights representation in both training and inference. They use a much larger minibatch size of 1000 and float32 backpropagation dataflow to accumulate more precise weight updates, equally compensating the buffer space in WAGE provided by external bits of $k_G$. However, large batch size will dramatically increase total training time, counteracting the speed benefits brought by integer arithmetic units. Besides, intermediate variables like feature maps often consume much more memory than weights and linearly correlated with mini-batch size. Therefore, we apply bigger $k_G$ for better convergence rate, accuracy and lower memory usage.

Table 4: Top-5 error rates (%) on ImageNet with different $k_G$ and $k_E$.

| Pattern | Vanilla | 28ff-BN | 28ff | 28f8 | 28C8 | 288C | 2888 |
|---------|---------|---------|------|------|------|------|------|
| error | 19.29 | 20.67 | 24.14 | 23.92 | 26.88 | 28.06 | 27.82 |

## 5 DISCUSSION AND FUTURE WORK

The goal of this work is to demonstrate potentials of applying training and inference with low-bitwidth integers in DNNs. Compared with FP16, 8-bit integer operations will not only reduce the energy and area costs for IC design (about $5\times$, see Table 5), but also halve the memory accesses costs and memory size requirements during training, which will greatly benefit mobile devices with on-site learning capability. There are some points not involved in this work but yet to be improved or solved in future algorithm developments and hardware deployment.

**MAC Operation**: WAGE framework is mainly tested with 2-8-8-8 bitwidth configuration, which means that though there are no multiplications during inference with ternary weights, MACs are still needed to calculate $g$ in training. Possible solution is 2-2-8-8 pattern if we do not consider the matching of bitwidths between $a$ and $e$. However, ternary $a$ will dramatically slow down convergence and hurt accuracy since $Q(x, 2)$ has two relatively high thresholds and clear most outputs of each layer at the beginning of training, this phenomenon is also observed in our BNN replication.

**Non-linear quantization**: The linear mapping with uniform distance is adopted in WAGE for its simplicity. However, non-linear quantization method like logarithmic representation (Miyashita et al., 2016; Zhou et al., 2017) might be more efficient because the weights and activations in a trained network naturally have logarithmic normal distributions as shown in Figure 4. Besides, values in logarithmic representation have much larger range with fewer bits than fixed-point representation and are naturally encoded in digital hardware. It is promising to training DNNs with integers encoded with logarithmic representation.

**Normalization**: Normalization layers like *Softmax* and batch normalization are avoided or removed in some WAGE demonstrations. We think normalizations are essential for end-to-end multi-channel perception where sensors with different modalities have different input distributions, as well as cross-model features encoding and cognition where information from different branches gather to form higher-level representations. Therefore, a better way to quantize normalization is of great interest in further studies.

Table 5: Rough relative costs in 45nm 0.9V from Sze et al. (2017).

| Operation | Energy(pJ) | | Area($\mu m^2$) | |
|-----------|------|------|------|------|
| | MUL | ADD | MUL | ADD |
| 8-bit INT | 0.2 pJ | 0.03 pJ | 282 | 36 |
| 16-bit FP | 1.1 pJ | 0.40 pJ | 1640 | 1360 |
| 32-bit FP | 3.7 pJ | 0.90 pJ | 7700 | 4184 |

## 6 CONCLUSION

WAGE empowers pure low-bitwidth integer dataflow in DNNs for both training and inference. We introduce a new initialization method and a layer-wise constant scaling factor to replace batch normalization, which is a pain spot for network quantization. Many other components for training are also considered or simplified by alternative solutions. In addition, the bitwidth requirements for error computation and gradient accumulation are explored. Experiments reveal that we can quantize relative values of gradients, as well as discard the majority of small values and their orders of magnitude in backpropagation. Although the accumulation for weights updates are indispensable for stable convergence and final accuracy, there still remain works for compression and memory consumption can be further reduced in training. WAGE achieves state-of-art accuracies on multiple

datasets with 2-8-8-8 bitwidth configuration. It is promising for incremental works via fine-tuning, more efficient mapping, quantization of batch normalization, etc. Overall, we introduce a framework without floating-point representation and demonstrate the potential to implement both discrete training and inference on integer-based lightweight ASIC or FPGA with on-site learning capability.

ACKNOWLEDGMENTS

This work is partially supported by the Project of NSFC (61327902), the SuZhou-Tsinghua innovation leading program (2016SZ0102), the National Natural Science Foundation of China (61603209) and the Independent Research Plan of Tsinghua University (20151080467). We discuss a lot with Peng Jiao and Lei Deng, gratefully acknowledge for their thoughtful comments.

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

# A  ALGORITHM

We assume that network structures are defined and initialized with Equation 5. The annotations after pseudo code are potential corresponding operations for implementation in a fixed-point dataflow.

---

**Algorithm 1** Training an $I$-layer net with WAGE method on floating-point-based or integer-based device. Weights, activations, gradients and errors are quantized according to Equations 6 - 12.

---

**Require:** a mini-batch of inputs and targets $(\boldsymbol{a}_q^0, \boldsymbol{a}^*)$ which are quantized to $k_A$-bit integers, shift-based $\alpha$ for each layer, learning rate scheduler $\eta$, previous weight $\boldsymbol{W}$ saved in $k_G$ bits.
**Ensure:** updated weights $\boldsymbol{W}_{t+1}$

    **1. Forward propagation:**

 1: **for** $i = 1$ to $I$ **do**
 2:    $\boldsymbol{W}_q^i \leftarrow Q_W(\boldsymbol{W}^i)$                             #Clip
 3:    $\boldsymbol{a}^i \leftarrow ReLU(\boldsymbol{a}_q^{i-1}\boldsymbol{W}_q^i)$                  #MAC, Clip
 4:    $\boldsymbol{a}_q^i \leftarrow Q_A(\boldsymbol{a}^i)$                          #Shift, Clip
 5: **end for**

    **2. Back propagation:**

    Compute $\boldsymbol{e}^I \leftarrow \frac{\partial \mathcal{L}}{\partial \boldsymbol{a}^I}$ knowing $\boldsymbol{a}^I$ and $\boldsymbol{a}^*$        #Substrate
 6: **for** $i = I$ to $1$ **do**
 7:    $\boldsymbol{e}_q^i \leftarrow Q_E(\boldsymbol{e}^i)$                        #Max, Shift, Clip
 8:    $\boldsymbol{e}^{i-1} \leftarrow \boldsymbol{e}_q^i\boldsymbol{W}_q^i$                  #MAC, Clip
 9:    $\boldsymbol{g}^i \leftarrow \boldsymbol{e}_q^{i\,\mathrm{T}}\boldsymbol{a}_q^{i-1}$                #MAC, Clip
10:    $\Delta \boldsymbol{W}^i \leftarrow Q_G(\boldsymbol{g}^i)$               #Max, Shift, Random, Clip
11:    Update and Clip $\boldsymbol{W}^i$ according to Equation 12    #Update, Clip
12: **end for**

---

# B LAYERWISE HISTOGRAM

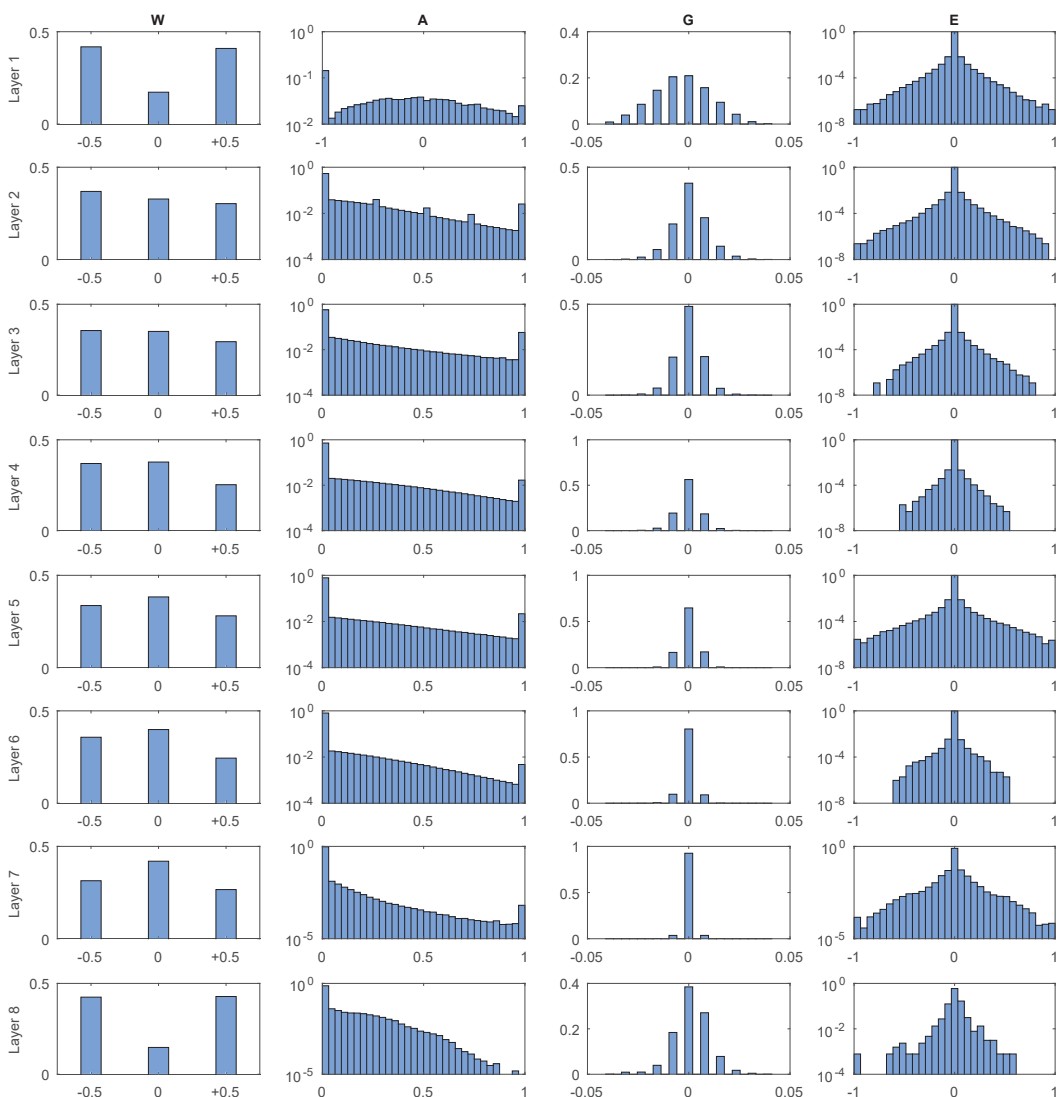

Figure 5: Layerwise histograms of a trained VGG-like network with bitwidth configuration: 2-8-8-8 and learning rate $\eta$ equals to 8. The Y-axis represents for probability in W-plots and G-plots, and logarithmic probability in A-plots and E-plots, respectively. In A-plots histograms are one-layer ahead so the first figure shows the quantized input data.

