# OpenReview forum: "Training and Inference with Integers in Deep Neural Networks"
_ICLR.cc/2018/Conference — Accept (Oral)_

### Official Review · AnonReviewer1 · 2017-11-24
**It is not clear if this work obtains significant improvements in comparison with previous works**

**Rating:** 7
**Confidence:** 4

**Review:**

This paper proposes a method to train neural networks with low precision. However, it is not clear if this work obtains significant improvements over previous works.

Note that:
1)	Working with 16bit, one can train neural networks with little to no reduction in performance. For example, on ImageNet with AlexNet one gets 45.11% top-1 error if we don’t do anything else, and 42.34% (similar to the 32-bit result) if we additionally adjust the loss scale (e.g., see Boris Ginsburg, Sergei Nikolaev, and Paulius Micikevicius. “Training of deep networks with halfprecision float.” NVidia GPU Technology Conference, 2017).
2)	ImageNet with AlexNet top-1 error (53.5%) in this paper seems rather high in comparison to previous works. Specifically, DoReFA and QNN, which used mostly lower precision (k_W=1, k_A=2 and k_E=6, k_G=32)  one can get much lower performance (47% and 49%, respectively). So, the main innovation here, in comparison, is k_G=12.
3)	Comparison using other datasets is made with different architectures then previous works, so it is hard to quantify what is the contribution of the proposed method. For example, on MNIST, the authors use a convolutional neural network, while BC and BNN used a fully connected neural network (the so called “permutation invariant mnist” problem).
4)	Cifar performance is good, but may seem less remarkable, given that “Gated XNOR Networks: Deep Neural Networks with Ternary Weights and Activations under a Unified Discretization Framework” already showed that k_G=k_W=k_A=2, k_E=32 is sufficient to get 7.5% error on CIFAR. So the main novelty, in comparison, is that k_E=12.

Taking all the above into account, it hard to be sure whether the proposed methods meaningfully improve existing methods. Moreover, I am not sure if decreasing the precision from 16bit to 12bit (as was done on ImageNet) is very useful for hardware applications, especially if there is such a degradation in accuracy. If, for example, the authors would have demonstrated all-8bit training on all datasets with little performance degradation, this would seem much more useful.

Minor: there are some typos that should be corrected, e.g.: “Empirically, We demonstrates” in abstract.

%%% Following the authors response %%%
The authors have improved their results and have addressed my concerns. I therefore raised my scores.

---

> ### Author Response · Authors · 2017-12-13
> **Reply to AnonReviewer1**
>
> We sincerely appreciate the reviewer for the comments, which indeed helps us to improve the quality of this paper.
>
> In our revised manuscript, we keep the last layer in full precision for ImageNet task (both BNN and DoReFa keep the first and the last layer in full precision). Our results have been improved from 53.5/28.6 with 28CC to 51.7/28.0 with 2888 bits setting. Results of other patterns are updated in Table4.  We have now revised the paper accordingly and would like to provide point-by-point response on how these comments have been addressed:
>
> (1) Working with 16bit, one can train neural networks with little to no reduction in performance.
>
> We introduce a thorough and flexible approach (from AnonReviewer3) towards training DNNs with fixed-point (8bit) integers, so there is no floating-point operands or operations in both inference and training phases. This is the key difference between our work and the previous works. As shown in Table5 in the revised manuscript, 5x reduction of energy and area costs can be achieved in this way, which we believe will greatly benefit the application of our method especially in mobile devices.
>
> (2) ImageNet with AlexNet top-1 error (53.5%) in this paper seems rather high in comparison to previous works.
>
> The significant differences between WAGE and existing works (DoReFa, QNN, BNN) lie in that:
>
>     1. WAGE does not need to store real-valued weights (DoReFa, QNN, BNN need).
>     2. WAGE calculates both gradients and errors with 8-bit integers (QNN, BNN use float32).
>     3. Many of the techniques, say for example, batch normalization and Adam optimizer that are hard to be
>          implemented on mobile devices are avoided by WAGE.
>
> Through experiments, we find that, if we store real-valued weights and do not quantize back propagation, the performance on ImageNet is at the same level (although not the same specification) as that of DoReFa, QNN and BNN.  Please refer to more detailed results in Table4.
>
> (3) Comparison using other datasets is made with different architectures then previous works
>
> Please refer to the comparison between TWN and WAGE in Table1 where we show a better result with the same CNN architecture.
>
> (4) Cifar performance is good, but may seem less remarkable.
>
> In fact,  k-E is set as 8 in WAGE. Gated-XNOR uses a batch size of 1000 and totally trains for 1000 epochs, so the total training time and memory consumption are unsatisfactory. Besides, they use float32 to calculate gradients and errors, and batch normalization layer is kept to guarantee the convergence.
>
> (5)  If, for example, the authors would have demonstrated all-8bit training on all datasets
>
> In our experiments, we find that it is necessary to set k-G>k-W, otherwise the updates of weights will directly influence the forward propagation and cause instability. Most of the previous works store real-valued weights (32-bits k-G), so they meet this restriction automatically. By considering this comment, we focus on 2-8-8-8 training and the results for ImageNet are updated in Table1 and Table4.

---

### Official Review · AnonReviewer2 · 2017-11-26
**The paper describe a method for how to train and make inference in a network using only integer values.**

**Rating:** 7
**Confidence:** 3

**Review:**

The authors describe a method called WAGE, which quantize all operands and operators in a neural network, specifically, the weights (W), activations (A), gradients (G), and errors (E) . The idea is using quantizers with clipping (denoted in the paper with Q(x,k)) and some additional operators like shift (denoted with shift(x)) and stochastic rounding. The main motivation of the authors in this work is to reduce the number of bits for representation in a network for all the WAGE operations and operands which influences the power consumption and silicon area in hardware implementations.

After introducing the idea and related work, the authors in Section 3 give details about how to perform the quantization. They introduce the additional operators needed for training in such network. Since quantization may loss some information, the authors need to quantize the signals in the network around the dynamic range in order not to "kill" the signal. The authors describe how to do that. Afterward, as in other techniques for quantization, they describe how to initialize the network values. Also, they argue that batch normalization in this network is replaced with the shift-quantize operations, and what is matter in this case is (1) the relative values (“orientations”) and not the absolute values and (2) small values in errors are negligible.

Afterward, the authors conduct experiments on MNIST, SVHN, CIFAR10, and ILSVRC12 datasets, where they show promising results compared to the errors provided by previous works. The WAGE parameters (i.e., the quantized no. of bits used) are 2-8-8-8, respectively. For understand more the WAGE, the authors compare on CIFAR10 the test error rate with vanilla CNN and show is small loss in using their network. The authors investigate mainly the bitwidth of errors and gradients.

In overall, this paper is an accept since it shows good performance on standard problems and invent some nice tricks to implement NN in hardware, for *both* training and inference. For inference only, other works has more to offer but this is a promising technique for learning. The things that are still missing in this work are some power reduction estimates as well as area reduction estimations. This will give the hardware community a clear vision of how such methods may be implemented both in data centers as well as on end portable devices.

---

> ### Author Response · Authors · 2017-12-13
> **Reply to AnonReviewer2**
>
> We thank the reviewer for the constructive suggestion:
>
> (1) The things that are still missing in this work are some power reduction estimates as well as area reduction estimations.
>
> We have taken this suggestion and added Table5 in Discussion, and made a rough estimation.
>
> For future work, we have tapped out our neuromorphic processors lately using phase-change memory to store weights and designed the ability to do some on-chip and on-site learning. The processor has 8-bit weights and 8-bit activation without any floating-point design. The real power consumption and area reduction of the processor has been simulated and estimated. It is very promising to implement some interesting application with continual learning demands on that chip as an end portable device.

---

### Official Review · AnonReviewer3 · 2017-11-28
**a thorough and flexible approach towards discretizing neural networks**

**Rating:** 8
**Confidence:** 4

**Review:**

The authors propose WAGE, which discretized weights, activations, gradients, and errors at both training and testing time. By quantization and shifting, SGD training without momentum, and removing the softmax at output layer as well, the model managed to remove all cumbersome computations from every aspect of the model, thus eliminating the need for a floating point unit completely. Moreover, by keeping up to 8-bit accuracy, the model performs even better than previously proposed models. I am eager to see a hardware realization for this method because of its promising results.

The model makes a unified discretization scheme for 4 different kinds of components, and the accuracy for each of the kind becomes independently adjustable. This makes the method quite flexible and has the potential to extend to more complicated networks, such as attention or memory.

One caveat is that there seem to be some conflictions in the results shown in Table 1, especially ImageNet. Given the number of bits each of the WAGE components asked for, a 28.5% top 5 error rate seems even lower than XNOR. I suspect it is due to the fact that gradients and errors need higher accuracy for real-valued input, but if that is the case, accuracies on SVHN and CIFAR-10 should also reflect that. Or, maybe it is due to hyperparameter setting or insufficient training time?

Also, dropout seems not conflicting with the discretization. If there are no other reasons, it would make sense to preserve the dropout in the network as well.

In general, the paper was written in good quality and in detail, I would recommend a clear accept.

---

> ### Author Response · Authors · 2017-12-13
> **Reply to AnonReviewer3**
>
> We thank the reviewer for the insightful comments. Please find our responses to individual questions below:
>
> (1) One caveat is that there seem to be some conflictions in the results shown in Table 1, especially ImageNet ...
>
> In our revised manuscript, we keep the last layer in full precision for ImageNet task (BNN and DoReFa kept both the first and the last layer), the accuracy for 2-8-8-8 is 51.7/28.0 compared to original results 53.5/28.6 with 2-8-C-C bits setting. Results of other patterns are updated in Table4.
>
> We find that the Softmax layer in the AlexNet model and 1000 categories jointly cause the conflictions. Since we make no exception for the first or the last layer, weights in the last layer will be limited to {-0.5,0,+0.5} and scaled by Equation(8), so the outputs of the last layer also obey a normal distribution N(0,1). The problem is that these values are small for a Softmax layer with 1000 categories.
>
> Example:
>
> x1=[0,0,0,…,1] (one-hot 1000 dims)
> y1=Softmax(x1)=[9.9e-4, 9.9e-4, …, 2.7e-3]
> e1 = z – x1, still a long way to train
> x2=[0, 0, 0,…,8] (one-hot 1000 dims)
> y2=Softmax(x2)=[1e-4, 1e-4, …, 0.75]
> e2 = z – x2, much closer to the label now
> label=z=[0,0,0,…,1].
>
> In this case, we observe that 80% weights in the last FC layer are trained greedily to {+0.5} to magnify the outputs. Therefore, the last layer would be a bottleneck for both inference and backpropagation. That might be why previous works do not quantize the last layer. The experiments on CIFAR10 and SVHN did not use Softmax cross-entropy and had only 10 categories, which indicates no accuracy drop.
>
>
> (2)Also, dropout seems not conflicting with the discretization...
>
> Yes, it is an additional method to alleviate over-fitting. Because we are working on designing a new neuromorphic computing chip, dropout will make the pipeline of weights and MAC calculations a little bit weird. Anyone who has no concern of that can easily add dropout to the WAGE graph.

---

### Public Comment · ~Tijmen_Blankevoort1 · 2018-02-05
**Similar paper + reference**

The idea of applying quantization this way on the weights, activations, gradients and errors was already discussed in Gupta et.al. 2015 (https://arxiv.org/abs/1502.02551). This paper applies stochastic rounding, in a fixed-point way on the above calculations/parameters to quantise a network while training. This paper does not appear in the reference, but given the similarity of the two papers, it's worthwhile including. It is also counter to the statement in this paper that this is the first bidirectional quantisation scheme that allows learning and inference in reduced bit-precision.
One of the things this paper does over Gupta et.al. 2015 is Distribution shifting. This is not novel, as it was already done in the flexpoint paper (Koster et.al. 2017) and before that in Courbariaux et. al. 2015 (https://arxiv.org/pdf/1412.7024.pdf)
I would like to see how this paper differs from the above mentioned papers.

---

> ### Author Response · Authors · 2018-02-10
> **Reply to Similar paper + reference**
>
> Thanks for your comments !
> --  I would like to see how this paper differs from the above mentioned papers.
> (1) In Gupta et.al. 2015 (https://arxiv.org/abs/1502.02551), they use WL = <IL + FL> to format the fixed-point parameters and intermediate variables. And in most of their experiments, the WL equals to 16, e.g., <6, 10>, <4, 12>. While in our paper, we only have IL part for sign bits, i.e., we use WL = 8 = <1 + 7>. In that case, we have to use distribution shifting. Compared to the half-precision floating-point (FP16), the computation efficiency and overhead of 16-bit fixed-point doesn't improve that much. However, we are considering to cite this paper and modify our wording.
> (2) Flexpoint (Koster et.al. 2017) use flexN+M to completely replace FP32 format in both training and inference, it addresses little about quantization. Besides, the distribution shifting in WAGE discard the exponent bits and only keeps the relatively orders of magnitude, which is quite different from the above papers.
> (3) Courbariaux et. al. 2015 (https://arxiv.org/pdf/1412.7024.pdf) use a higher precision for the parameters during the updates than during the forward and backward propagations, which can really improve the performance. Please see http://papers.nips.cc/paper/7163-training-quantized-nets-a-deeper-understanding.

---

### Public Comment · ~Rudra_Poudel1 · 2018-02-13
**Gradient of quantitized integer values/variables**

How do you calculate the gradient of (quantized/discontinuous) integer weight (W_{q}^i) and activation (e_{q}^i)?

---

> ### Author Response · Authors · 2018-02-13
> **Reply to Gradient of quantitized integer values/variables**
>
> Please refer to the algorithm in Appendix A, we use back-propagation and quantize deltaW and e_{q}^i within each layer.

---

### Public Comment · ~herbert_chen1 · 2018-03-28
**have you try (mod P) congruence class, which is a typical field**

maybe it can change  classification to regression，1/2=2 mod（3），1/1=1 mod（3）。with some Convolution ，it seems interesting

---

### Decision · Program_Chairs · 2018-01-29
**ICLR 2018 Conference Acceptance Decision**

**Decision:**

Accept (Oral)

**Comment:**

High quality paper, appreciated by reviewers, likely to be of substantial interest to the community. It's worth an oral to facilitate a group discussion.